# Unconscious Conflict Adaptation of Heroin Abstainers

**DOI:** 10.3390/jcm11216504

**Published:** 2022-11-02

**Authors:** Ling Li, Changhu Yan, Hua Cao, Ling Yang, Yuchen Luo, Yu Zhao, Xiao Lu

**Affiliations:** School of Psychology, Northwest Normal University, Lanzhou 730071, China

**Keywords:** conflict adaptation, unconscious conflict adaptation, heroin abstainers

## Abstract

Conflict adaptation is representative of the dynamic cognitive control process, which reflects the adaptability and flexibility of personal cognitive processing. Cognitive control plays an important role in drug use and relapse in addicts. Previous studies have identified conscious conflict adaptation in drug addicts. The present study examined unconscious conflict adaptation in persons with heroin use disorder using an arrow version meta-contrast masking task. The results found that persons with heroin use disorder had smaller unconscious conflict adaptation compared to the healthy control group. This may be a result of functional brain damage caused by long-term drug use.

## 1. Introduction

People frequently draw on their previous experiences to aid themselves in completing the current activities in regular living. According to the ancient Chinese proverb, “a fall into the pit, a gain in your wit,” when one confronts a difficulty, wisdom grows and experiences are gained. In a similar manner, people who have dealt with a certain form of conflict before may draw on that experience when confronted with a conflict of a similar nature in the future. In the field of cognitive control, this phenomenon is known as conflict adaptation, and is manifested by the fact that the conflict experienced in the previous trial results in faster and more accurate responses in the current trial. Conflict adaptation was originally noted in the Flanker task by Gratton et al. [1], where the congruence effect following inconsistent trials was less significant than the consistency effect following consistent trials. Later researchers noticed conflict adaptation in the Simon task and Stroop task [2,3,4,5,6]. Conflict adaptation reflects not only the influence of prior experience on current behavior, but also the flexibility and adaptability of human cognitive processing [7,8].

Drug addiction is a chronic relapsing disorder characterized by persistent brain changes associated with cognitive, motivational, and affective alterations [9]. Imaging studies demonstrate that drug addictions have an impact on the brain’s cognitive control function [10,11]. Cognitive control impairments have also been observed in drug addicts in tasks requiring suppression of task-irrelevant information [12], attention bias [13], and cognitive control [14]. Impaired cognitive control is associated with relapse in addicts [15]. The detection of individual conflict adaptation can effectively reflect their cognitive control function [16]. Therefore, it is very meaningful to study the conflict adaptation of drug addicts. Studies have found that there are defects in conflict adaptation in drug addicts [16,17]. Zhou et al. [17] used ERP technology to explore the conflict adaptation effect of long-term heroin addiction using the letter flanking task. The results suggested that the heroin addiction group showed conflict adaptation effects on behavioral outcomes, but N2 amplitude did not differ among different trial types. The conflict monitoring theory provides an explanation for this result [18]. According to the conflict monitoring theory, individuals’ monitoring of previous conflicts enhances their ability to exercise better cognitive control and adjust to accomplish the task in the current trial. Specifically, when the previous trial was congruent, the level of cognitive control was relatively low because there was no conflict information to trigger the adjustment mechanism; in contrast, when the previous trial was incongruent, the adjustment mechanism was triggered and the level of cognitive control was relatively high, which produced a smaller consistency effect following incongruent trials than following congruent trials, as an “adaptation” [19,20]. In terms of neural mechanisms, the conflict monitoring theory postulates that the conflict in the previous trial activates the anterior ungulate cortex (ACC), which then signals the monitored conflict to the dorsal lateral prefrontal cortex (DLPFC), a brain region specifically in charge of control, so that the brain is in an active state of readiness before the current trial and is better able to control the conflict after the current trial has taken place [21]. In the research of Zhou et al., participants allocated more attention control resources after experiencing high conflict trials (i.e., I trials) to reduce conflict activities, resulting in a reduction of N2 amplitude, which suggested a reduction in task-independent information processing and conflict activation. Therefore, N2 effect did appear in the healthy control group, but long-term drug abuse would lead to the impairment of ACC and DLPFC function in drug addicts [22], so N2 effect disappeared in the heroin addiction group.

Previous views have suggested that cognitive control is conscious dependent, because previous studies have found that the arousal and execution of cognitive control depend on the function of the prefrontal cortex, which in turn is associated with conscious experience [19,20,23], and thus researchers naturally associated cognitive control with consciousness. However, in recent years, several studies have found that unconscious information can also trigger cognitive control, such as inhibitory control [24,25]. Conflict adaptation can similarly be activated by both conscious and unconscious information [8,26]. Unconscious inputs initiated the conflict adaptation process and led to error and conflict behavior adjustment [27]. Both behavioral and cognitive neural studies have also revealed the presence of unconscious conflict adaptation [8,26,27,28,29]. For example, Jiang et al. used an arrow version meta-contrast masking task and found significant conflict adaptation effects in both conscious and unconscious conditions at response time and in frontal N2 and mid-parietal P3 amplitudes [8]. Similar results have been obtained by other researchers [30].

The aim of this study was to explore unconscious conflict adaptation in drug addicts. Deficits in conflict adaptation in drug addicts have been confirmed by relevant studies [16,17]. Other studies have also demonstrated that drug addicts are able to process unconscious drug-related cues as well as emotion-related information [31,32], implying that drug addicts are inevitably influenced by “unseen” information. Therefore, the conflict adaptation induced by unconscious stimuli in drug addicts became the focus of our study. For experimental purposes, the present study also used an arrow version meta-contrast masking task [8,26]. In this task, the priming stimuli had a similar perceptual profile to the target stimuli, and thus, when it was shown for a shorter period of time, participants typically were not aware of its presence, accomplishing masking. Previous studies have found early conflict monitoring impairment and late response conflict resolution processing abnormalities in drug addicts [33]. Therefore, we hypothesized that drug addicts had impaired unconscious conflict adaptation.

## 2. Materials and Methods

### 2.1. Participants

Thirty-eight male heroin abstainers who were receiving compulsory isolated abstinence treatment from the Addiction Recovery Center of Gansu Province, and 36 healthy males without a history of drug abuse who were matched in age and education level, through community advertisement were recruited to participate in this experiment. The following inclusion criteria were used for the heroin abstainer group: (1) met the criteria for opioid users in the Diagnostic and Statistical Manual of Mental Disorders (4th ed., DSM-IV), as assessed by professional physicians in the Addiction Recovery Center; (2) addicted to heroin only; (3) normal vision or corrected vision; (4) no history of mental illness or other serious diseases. The inclusion criteria for the healthy control group were: (1) normal vision or corrected vision; (2) no history of mental illness and other serious diseases. The mean age of the heroin abstainer group was 48.58 ± 4.53 years, the first drug use age was 30.24 ± 9.70 years, and the mean duration of current abstinence was 8.42 ± 4.62 months (range: 4–24 months). The mean age of the healthy control group was 50.33 ± 5.21 years. There was no significant difference in age between the two groups (*t* = 1.546, *p* > 0.05). All participants signed informed consent prior to the experiment, and this study was approved by the Ethics Committee of the School.

### 2.2. Experimental Design

A three-factor mixed experimental design of 2 (group: heroin abstainer group, healthy control group) × 2 (Previous-trial Congruence: congruent, incongruent) × 2 (Current-trial Congruence: congruent, incongruent) was used. Among them, the group was the between-subject variable, and the previous-trial congruence and the current-trial congruence were the within-subject variables. The dependent variables were the response time and error rate of the participants to the target stimuli.

Based on the previous-trial congruence and the current-trial congruence, the types of trial combinations can be divided into four categories: congruent trials following congruent trials (cC), incongruent trials following congruent trials (iC), congruent trials following incongruent trials (cI), and incongruent trials following incongruent trials (iI). Conflict adaptation is represented if participants statistically show an interaction between previous-trial congruence and current-trial congruence. In terms of reaction times or error rates, the magnitude of conflict adaptation can be calculated by the formula: conflict adaptation = RT [(cI-cC) − (iI-iC)]; the larger the value, the stronger the conflict adaptation effect [34,35].

### 2.3. Experimental Process

The arrow version meta-contrast masking task, which has been widely used in the field of unconscious conflict adaptation, was adopted [8,26]. Stimuli were displayed on a 24-inch monitor with a refresh rate of 60 Hz and a resolution of 1024 × 768. Participants were 70 cm away from the screen. The E-prime 2.0 software package was used to present stimuli and timing. A hollow black arrow pointing left or right (width 0.82°, height 0.41°) was used for the priming stimuli, and a hollow black arrow pointing left or right (width 2.45°, height 1.23°) and slightly larger than the priming arrow was used for the target stimuli.

In the formal experiment, firstly, the priming arrow was presented for 17 ms, followed by a blank interval for 33 ms, then the target arrow was presented for 233 ms, and finally a random blank interval with a time interval between 1200 and 1500 ms was presented (see Figure 1). Participants were asked to respond quickly and accurately to the direction of the target stimulus by pressing the “F” key for the left arrow and the “J” key for the right arrow. The formal experiment consisted of seven blocks, each with 128 trials, with an equal number of trials for each category of trial combination. Prior to the formal experiment, participants were asked to perform a practice block with 24 trials.

After the formal experiment, participants were also asked to complete a two-alternative forced-choice discrimination task to determine whether participants were able to perceive the orientation of the masked prime arrow. The task consisted of 128 trials, each starting with a priming arrow (17 ms), followed by a blank interval (33 ms), then a target arrow (233 ms), and finally a random blank interval (1000–1500 ms). Participants were required to judge the orientation of the priming arrow. There was no time limit for the reaction. After the participant presses the button, the option disappears and the next trial begins. In addition, the heroin abstainer group was required to complete a Visual Analog Scale (from not at all to strong craving) before and after the experiment to assess subjective drug craving. Both the heroin abstainer group and the healthy control group were required to complete a nine-point rating of emotional state before and after completing the experiment.

### 2.4. Data Analysis

Firstly, we used a two-alternative forced-choice discrimination task to verify the effectiveness of the subliminal presentation of stimuli. To explore participants’ awareness of the masked priming stimuli, we performed a one-sample *t*-test on participants’ responses versus chance-level performance [26]. Secondly, the first trial of each block, and the error trials and the first correct trial following the error trial [8], as well as trials with a reaction time greater than 1000 ms or less than 100 ms, were excluded from data analysis [26]. To explore the effects of conflict adaptation in the heroin abstainer group and the healthy control group, we performed a three-factor repeated measures analysis of variance for response time and error rates by group (heroin abstainer group, healthy control group) × Previous-trial Congruence (congruent, incongruent) × Current-trial Congruence (congruent, incongruent). Finally, according to the formula of conflict adaptation (Conflict effect = RT [(cI-cC) − (iI-iC)]) [34], we calculated the conflict adaptation values for the abstainer group and the healthy control group, respectively. The conflict adaptation of the two groups was tested by the independent-sample *T*-test.

## 3. Results

### 3.1. Awareness of Primes

All participants reported not being able to perceive the masked primes prior to the targets. To test whether participants actually perceived the masked priming stimuli, a one-sample *T*-test was used to test the difference between the correct rate of response to priming stimuli and the chance-level for the participants in the two groups. Results showed that the heroin abstainer group (*t*(37) = 1.58, *p* > 0.05) and the healthy control group (*t*(35) = 1.46, *p* > 0.05) could not consciously perceive the presence of the masked priming stimuli, indicating that the manipulation of the masked prime in the experiment was effective.

### 3.2. Emotional States and Drug Craving

Emotional states affect individuals’ conflict adaptation [36,37]. Therefore, a repeated measures ANOVA on the pre-experimental and post-experimental emotional states of the two groups showed that the difference between the pre-test and post-test of emotional state was not significant, *F*(1, 72) = 0.15, *p* > 0.05, and the interaction between pre- and post-measures and group was also not significant, *F*(1, 72) = 0.29, *p* > 0.05. This suggested that the results of this study were not influenced by the emotional states of the participants.

Craving affects the cognitive functioning of addicts [38]. Therefore, a paired- samples *T*-test on the craving of the heroin abstainer group revealed a non-significant difference between the craving before and after the experiment, *t*(37) = 0.66, *p* > 0.05, indicating that the results of this study were not affected by the drug craving of the heroin abstainer group.

### 3.3. Conflict Adaptation Reflected in RTs and Error Rates

The reaction times (RTs) and error rates on the flanker task were analyzed by three factor repeated measures analysis of variance of group (heroin abstainer group, healthy control group) × Previous-trial Congruence (congruent, incongruent) × Current-trial Congruence (congruent, incongruent). Mean RTs and error rates of all factorial combinations of the included variables are presented in Table 1 and Table 2.

The results of the RTs indicated that the main effect of group was not significant (*F*(1, 72) = 0.21, *p* > 0.05) and the main effect of previous-trial congruence was not significant (*F*(1, 72) = 0.41, *p* > 0.05), but the main effect of current-trial congruence was significant (*F*(1, 72) = 85.41, *p* < 0.001, *η*_p_^2^ = 0.543), showing that the RTs of incongruent trials were significantly higher than those of congruent trials. The interaction of three factors was significant (*F*(1, 72) = 5.36, *p* < 0.05, *η*_p_^2^ = 0.069). Further analysis showed that in the heroin abstainer group, the main effect of previous-trial congruence was significant (*F*(1, 37) = 13.80, *p* < 0.01, *η*_p_^2^ = 0.272), showing that the RTs of incongruent trials were significantly higher than those of congruent trials. The main effect of current-trial congruence was significant (*F*(1, 37) = 34.56, *p* < 0.001, *η*_p_^2^ = 0.483), showing that the RTs of incongruent trials were significantly higher than those of congruent trials. The interaction between previous-trial congruence and current-trial congruence was significant (*F*(1, 37) = 10.97, *p* < 0.01, *η*_p_^2^ = 0.229), showing that the conflict effect after incongruent trials (*M* = 12.90 ms) was significantly smaller than that after congruent trials (*M* = 28.09 ms, *t*(37) = −3.31, *p* < 0.01). This suggested conflict adaptation in RTs in the heroin abstainer group. In the healthy control group, the main effect of previous-trial congruence was not significant (*F*(1, 35) = 2.62, *p* > 0.05). The main effect of current-trial congruence was significant (*F*(1, 35) = 51.35, *p* < 0.001, *η*_p_^2^ = 0.595), showing that the RTs of incongruent trials were significantly higher than those of congruent trials. The interaction between previous-trial congruence and current-trial congruence was significant (*F*(1, 35) = 23.04, *p* < 0.001, *η*_p_^2^ = 0.397), showing that the conflict effect after incongruent trials (*M* = 8.50 ms) was significantly smaller than that after congruent trials (*M* = 43.35 ms, *t*(37) = −4.80, *p* < 0.001). This suggested conflict adaptation in RTs in the healthy control group. To further compare the conflict adaptation between the heroin abstainer group and the healthy control group, we used the independent-sample *T*-test. The results showed that the conflict adaptation of the heroin abstainer group (*M* = 15.19 ms) was significantly smaller than that of the healthy control group (*M* = 34.85 ms, *t*(35) = −2.31, *p* < 0.05).

The results of error rates indicated that the main effect of group was not significant (*F*(1, 72) = 1.01, *p* > 0.05) and the main effect of previous-trial congruence was not significant (*F*(1, 72) = 1.08, *p* > 0.05), but the main effect of current-trial congruence was significant (*F*(1, 72) = 93.51, *p* < 0.001, *η*_p_^2^ = 0.565), showing that the error rates of incongruent trials were significantly higher than those of congruent trials. The interaction of three factors was not significant (*F*(1, 72) = 3.72, *p* > 0.05). The interaction between previous-trial congruence and current-trial congruence was significant (*F*(1, 72) = 85.53, *p* < 0.001, *η*_p_^2^ = 0.543). Further analysis showed that in the heroin abstainer group, the interaction between previous-trial congruence and current-trial congruence was significant (*F*(1, 37) = 76.47, *p* < 0.001, *η*_p_^2^ = 0.674), showing that the conflict effect after incongruent trials (*M* = 0.01) was significantly smaller than that after congruent trials (*M* = 0.10, *t*(37) = −8.75, *p* < 0.001). This suggested conflict adaptation in error rates in the heroin abstainer group. In the healthy control group, the interaction between previous-trial congruence and current-trial concurrency was significant (*F*(1, 35) = 22.30, *p* < 0.001, *η*_p_^2^ = 0.389), showing that the conflict effect after incongruent trials (*M* = 0.02) was significantly smaller than that after congruent trials (*M* = 0.09, *t*(37) = −4.72, *p* < 0.001). This suggested conflict adaptation in error rates in the healthy control group.

## 4. Discussion

By manipulating the timing of presentation of the priming stimuli, the present study explored the unconscious conflict adaptation effect in heroin addicts using an arrow version meta-contrast masking task. The findings showed that both the heroin abstainer group and the healthy control groups exhibited significant conflict adaptation effects in the unconscious condition. More crucially, the conflict adaptation was significantly lower in the heroin abstainer group than in the healthy control group, indicating impairment in conflict adaptation. This was in line with our experimental hypothesis.

Firstly, the results of the overall analysis revealed an interaction between previous-trial congruence and current-trial congruence, and the conflict effect following incongruent trials was significantly smaller than the conflict adaptation following congruent trials, i.e., an unconscious conflict adaptation effect emerged. This replicated the results of conscious conflict adaptation, suggesting that unconscious conflict can also induce adaptive behavior in individuals as well as conscious conflict, consistent with previous research findings [8,26,39]. It might also be an indication that individuals actively strengthen cognitive control to prevent misbehavior from repeating once more or to adjust their current conflict behaviors after being exposed to unconscious conflict-evoking stimuli. According to conflict monitoring theory [19,20], the conflict in the previous incongruent trial would be identified by the conflict monitoring system, which then transmits the conflict information to the conflict control system, and the activated dorsolateral prefrontal cortex (DLPFC) then deploys more cognitive resources to better control the conflict in the current trial through top-down regulation, improving the participants’ performance in the current incongruent trials. Thus, the phenomenon of unconscious conflict adaptation emerged. This also illustrated how adaptable and flexible human cognitive processing is [7]. As confirmed by earlier research, conscious conflict adaptation outperforms unconscious conflict adaptation [26,30,40]. According to the view that the difference between unconscious and conscious conflict control is quantitative rather than an all-or-none phenomenon [40], this may indicate variations in the degree of activation of the cognitive control system by unconscious and conscious conflict [8]. However, as the present study did not further investigate conscious conflict adaptation, no additional inferences would be drawn.

More notably, compared to the healthy control group, the present study found that unconscious conflict adaptation was much smaller in persons with heroin use disorder. It is possible that this was the first study of unconscious conflict adaptation in drug users. Prior studies on persons with substance use disorder for conscious conflict adaptation have revealed that these individuals had impaired conscious conflict adaptation [11,17]. The results of the present study concur with Solo’s findings [16]. This could also be a sign that persons with substance use disorder have impaired conflict adaptation on both the conscious and unconscious levels. This might be as a result of the functional brain damage brought on by chronic heroin use. For instance, research has shown that heroin substances use disorder leads to reduced ACC activity [41]. Forman also observed error-related decreased ACC activity and significantly poor task performance among opiate users [42]. The frontal lobe is particularly vulnerable to the acute and chronic effects of addictive drugs, and a drug-induced hyperdopaminergic condition causes widespread hypoactivity of the prefrontal cortex, including the orbitofrontal, dorsolateral, and medial regions [43]. A significant relationship between trial-to-trial behavioral RTs and PFC activity was seen in persons with drug use disorder, adding evidence that a PFC dysfunction underpins the deficit in conflict adaptation of persons with drug use disorder [16]. Conflict monitoring theory suggests that the anterior cingulate cortex (ACC) and the dorsolateral prefrontal lobe (DLPFC) are crucial for conflict adaptation [19]. Extensive fMRI studies confirmed the role of the ACC in monitoring response conflict, and the PFC in managing the conflict [20,21,44]. Additionally, conflict adaptation reflects the cognitive flexibility of individual processing [26]. Some studies found that persons with drug use disorder performed cognitive flexibility tasks with much longer response times and more errors than the control group, demonstrating a lack of cognitive flexibility [45,46]. All these provide evidence to support our speculation. Therefore, it is reasonable to suspect that abnormal conflict adaptation in persons with heroin use disorder results from impaired cognitive function induced by long-term drug use.

## 5. Conclusions

An arrow version meta-contrast masking task was used in the present study to explore unconscious conflict adaptation in persons with heroin use disorder. According to the findings, heroin users exhibited much lower conflict effects following incongruent trials than they did following congruent trials, indicating that an unconscious conflict adaptation effect had occurred. More significantly, heroin substances use disorder’s unconscious conflict adaptation effect was significantly smaller than that in the healthy control group, indicating that persons with heroin use disorder may have impaired unconscious conflict adaptation as a result of functional brain impairment brought on by long-term drug use.

## Figures and Tables

**Figure 1 jcm-11-06504-f001:**
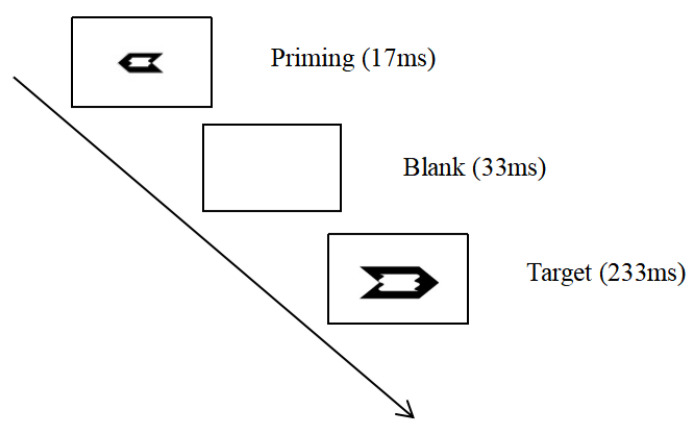
Experimental process.

**Table 1 jcm-11-06504-t001:** Conflict adaptation results reflected in RTs.

	Heroin Abstainer Group		Healthy Control Group	
	Current-Congruent	Current-Incongruent	Current-Congruent	Current-Incongruent
Previous-congruent	434.26 ± 77.56	462.35 ± 84.18	437.08 ± 95.08	480.43 ± 110.56
Previous-incongruent	464.99 ± 79.37	477.89 ± 76.74	438.68 ± 97.38	447.18 ± 91.60

Note: RTs = [(cI-cC) − (iI-iC)].

**Table 2 jcm-11-06504-t002:** Conflict adaptation results reflected in error rates.

	Heroin Abstainer Group		Healthy Control Group	
	Current-Congruent	Current-Incongruent	Current-Congruent	Current-Incongruent
Previous-congruent	0.26 ± 0.12	0.36 ± 0.13	0.34 ± 0.15	0.42 ± 0.14
Previous-incongruent	0.36 ± 0.15	0.38 ± 0.16	0.35 ± 0.15	0.36 ± 0.15

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
