# Peer review of "Unconscious Conflict Adaptation of Heroin Abstainers"

_jcm, 2022, doi:10.3390/jcm11216504_

Round 1

Reviewer 1 Report

This is a potentially interesting study. The aim is to examine “Unconscious Conflict Adaptation of Heroin Abstainers”. The basic premise is well known-i.e. that drug users/abstainers have some kind of cognitive deficit that impairs decision making. However what is less clear is what new information this study provides. Also what the implications of this deficit if it exists. Is it temporary or permanent?  Is there some threshold in terms of the impairment in terms of drug related behaviour?  Also the covariates mentioned in the text. I appreciate this is a complex topic and would welcome some further reflection on these topics.  The paper also needs  a hypothesis/analysis section.

Line Number/Comment

9          Abnormal inhibition and control is an essential factor for the development and persistence of heroin addiction”. Perhaps in some cases but is it necessary?

13        examines the unconscious conflict adaptation of heroin abstainers by using arrows in a partial contrast masking task. What is the new feature of this task that has not been examined?

14        “the conflict adaptation value of the heroin abstainers group was significantly less than that of the health control group What are the implications- this a fixed factor or something that can be modified? Especially since these were abstainers. (How long had they been abstainers)? This kind of study is well known-so what new information does this study provide?

27. “impulsive drug-seeking behavior and relapse behavior are the core symptoms in the withdrawal period”. Just this period or  before as well?

35 Explain flankers task. Next paragraph is quite general. Presumably there is individual variation?

62        “This can help to explain the ability of addicts to suppress unconscious stimuli, and further reveal the relationship between the anti-interference ability and relapse behavior of heroin abstainers when facing drug-related stimuli.” I think the introduction could be more specific on this point.

71        So  these are long time drug users/addicts?

81        Impulsivity can be assessed from questionnaire, what does this new task add? How does this correlate with experimental results?

147. The paper has  no analysis section-this makes the results difficult to interpret. There should be a statement of all he experimental hypotheses.

150 pre-test and post-test of the heroin abstainers group’s craving for drugs (t(37)= 0.66, p>.005. Was this anticipated? The whole results section is complicated and difficult to follow. Table 1 is not referred to in the text.

213. Again, what is the specifically new finding here?

225. Was the amount of conflict especially large?

197      From here to 209. These results are interesting-what do they mean? What about the pre study measure of drug craving/impulsivity?

213      Discuss absolute value in more detail-for example, is there a threshold? Can we say that over a certain level is going to make withdrawal harder.

235. “This may indicate that the conflict inhibition function of drug addicts is impaired due to long-term drug abuse, making addicts spend more time in solving conflict inhibition than health people[31]. In addition, in a study of electroencephalogram” does this deficit exist prior to drug  or is it the result of drug use. Note: y missing for health.

247      In the face of conflict, more cognitive resources need to be mobilized, and the control of irrelevant information interference was poor.” Not sure what this means. If the conflict is not conscious how can “more cognitive resources need to be mobilized”. What are these cognitive resources?

264-271. Does this mean that it is depression rather than drug use that is related to poorer performance?

288 “cognitive control ability of unconscious conflict improved with abstinence time” Earlier text does not mention this?

303.  The conclusion could really draw out what new information this study provides.

Author Response

Reviewer1:

This is a potentially interesting study. The aim is to examine “Unconscious Conflict Adaptation of Heroin Abstainers”. The basic premise is well known-i.e. that drug users/abstainers have some kind of cognitive deficit that impairs decision making. However what is less clear is what new information this study provides. Also what the implications of this deficit if it exists. Is it temporary or permanent?  Is there some threshold in terms of the impairment in terms of drug related behaviour?  Also the covariates mentioned in the text. I appreciate this is a complex topic and would welcome some further reflection on these topics.  The paper also needs  a hypothesis/analysis section.

Answer:First of all, thank the experts for their valuable opinions. Inspired by experts, we reorganized the research logic and deeply thought about the research problems. Previous studies have found that the cognitive control function of addicts is closely related to their relapse. Conflict adaptation, as an effective indicator of the cognitive control function of addicts, reflects the cognitive control and cognitive flexibility of individuals. Previous studies have found that drug addicts' conflict adaptation is impaired. As some studies have found that the cognitive control function of addicts is closely related to their relapse, conflict adaptation, as an effective indicator to reflect the cognitive control function of addicts, will naturally have an impact on the social adaptation and relapse of addicts.

Addicts' conflict adaptation has been proved to be damaged at the conscious level, so there are a lot of unconscious information in the process of addicts' withdrawal, which will also have an impact on their withdrawal, such as subliminal drug-related clues, stigmatization of addicts and other unconscious information. When addicts face the interference of such unconscious information, what characteristics will their conflict adaptation have become the focus of our research. We assume that heroin addicts are in unconscious conflict adaptation and accompanied by injury. Our research results finally found that addicts' unconscious conflict adaptation is impaired, that is to say, heroin addicts' anti-interference ability will be reduced when faced with unconscious information interference.

As for the addiction's conflict adjustment damage you mentioned, we are not sure whether it is temporary or permanent. As mentioned earlier, long-term drug abuse leads to the impairment of cognitive control function of heroin addicts, and many researchers have tried to study whether this loss can be recovered through drug withdrawal. Li et al. pointed out that the functional impairment caused by drug abuse is an irreversible and persistent brain injury. However, Wang et al. found that the density of gray matter in the frontal gyrus of heroin addicts who quit for one month returned to normal level. The follow-up study on cocaine abstainers also pointed out that the activation level of each node in the reaction inhibition circuit was enhanced and there was no significant difference with the normal group. However, Zhou's research on the inhibition of suprathreshold response suggests that the conflict adaptation of long-term abstainers still has damage, so we think that this damage may take longer to repair, or it may not be repaired. Of course, we are also considering conducting a follow-up study to explore whether this conflict adaptation of addicts is reversible.

As for the question mentioned by experts as to whether there is a threshold, previous researchers have confirmed that the conflict adaptation of addicts has damage at the suprathreshold level of consciousness. Our study is to discuss whether there is conflict adaptation and damage in the subthreshold unconscious information of addicts. Specifically, we operate the subthreshold and subthreshold by manipulating the presentation time or masking. When the stimulus is 17ms, it is the subthreshold unconscious stimulus. Or your question refers to whether there is a threshold for the conflict adaptation impairment of addicts. The definition of impairment in previous studies was generated by comparison with the control group (healthy population). When the cognitive performance of addicts was worse than that of the control group, they presented as impairment, and there was no specific threshold.

In the last part about covariates and research hypotheses mentioned by experts, we found that negative emotions do affect individuals' conflict adaptation after consulting relevant literature. In our research, negative emotions have no significant impact on the experimental group, but the control group has significant impact. We also explained the results in the discussion part, and we think that negative emotions have different effects on the conflict adaptation of the two groups of subjects (see the third paragraph of the discussion for details). Finally, we re-add the research hypothesis in the introduction. We assume that heroin addicts are in unconscious conflict adaptation and accompanied by injury. (See the last paragraph of the introduction for details).

Question 1:Abnormal inhibition and control is an essential factor for the development and persistence of heroin addiction”. Perhaps in some cases but is it necessary?

Answer:There have been many factors that contribute to addiction as well as relapse, however, research has found that cognitive impairment is an important factor that leads addicts to control their craving urges and therefore relapse, therefore, abnormal cognitive control is an important factor in the persistence and relapse of drug addicts. As we have written:Drug addiction is a chronic relapsing disorder characterized by persistent brain changes associated with cognitive, motivational, and affective alterations. Imaging studies demonstrate that drug addiction have an impact on the brain's cognitive control function. Cognitive control impairments have also been observed in drug addicts in tasks requiring suppression of task-irrelevant information, attentional bias and cognitive control. Impaired cognitive control is associated with relapse in addicts.

Question2:examines the unconscious conflict adaptation of heroin abstainers by using arrows in a partial contrast masking task. What is the new feature of this task that has not been examined?

Answer:Thanks for the expert's correction, this part has been revised again in the text. This task is a classic paradigm for the study of unconscious conflict adaptation. Researchers use this task when conflict adaptation is first discovered. Flanker task can not only test the ability of subjects to identify and resolve conflicts in a single trial, but also test the ability of individuals to adjust conflict trials under the background of conflicts. Therefore, it is widely used in the related research of conflict adaptation in the future. Because our masking task needs to check whether subliminal manipulation is effective, the final result shows that subliminal manipulation is effective, which proves the effectiveness of our experimental task.

Question3:“the conflict adaptation value of the heroin abstainers group was significantly less than that of the health control group What are the implications- this a fixed factor or something that can be modified? Especially since these were abstainers. (How long had they been abstainers)? This kind of study is well known-so what new information does this study provide?

Answer:Thanks for the expert's opinion. The results show that the unconscious conflict adaptation of addicts is impaired. That is to say, when heroin addicts face unconscious information interference, their anti-interference ability will also be reduced, and then there may be relapse and other behaviors. As for the problem that you mentioned that the addiction's conflict adaptation damage is temporary or permanent, we think that we can also consider conducting a follow-up study in the future to explore whether the addiction's conflict adaptation is reversible. The average abstinence time of these heroin abstainers is two years. Previous studies on the conflict adaptation of addicts found that there was no significant difference in behavioral response between them and non-addicts, but there were significant differences in EEG indicators, and there were obvious defects in the conflict adaptation of addicts. Our research on unconscious conflict adaptation shows that there are obvious differences in behavioral responses between them, which probably indicates that there are different mechanisms of conflict adaptation between conscious level and unconscious level, and unconscious information has more obvious interference effect on addicts' conflict adaptation.

Question4:“impulsive drug-seeking behavior and relapse behavior are the core symptoms in the withdrawal period”. Just this period or  before as well?

Answer:Thank you for the expert's warning and correction. Once again, we apologize to the experts for the mistakes in our expression. In order to make the article concise and logical, the content has been deleted.

Question5:Explain flankers task. Next paragraph is quite general. Presumably there is individual variation?

Answer:Thanks for the expert's reminder. In the arrow Flanker task, the subjects were presented with a row of five arrows, and asked to respond to the target arrow in the middle and ignore the interfering arrows on both sides. Set two types of stimuli: consistent (C, interference arrow is consistent with target arrow) and inconsistent (I, interference arrow is inconsistent with target arrow). Usually, the subjects' response to consistent attempts is faster than that to inconsistent attempts, and the difference of the reaction time between the two attempts is the flanker interference effect. If the relationship between before and after trials is also included in the analysis, there will be four types of stimulation: CC (previously consistent and currently consistent), CI (previously consistent and currently inconsistent), IC (previously inconsistent and currently consistent) and II (previously inconsistent and currently inconsistent). The adaptive effect means that the interference effect (II-IC) after the inconsistent trial is smaller than the interference effect (CI-CC) after the consistent trial, that is, (CI-CC)>(II-IC), and the magnitude of the adaptive effect is expressed by [(CI-CC)-(II-IC)]. Researchers generally believe that conflict adaptation effect reflects the ability of subjects to make dynamic adjustments in control. As for the question whether there are individual differences in conflict adaptation, Chinese scholar Wang Ting combined various neuroscience techniques in her doctoral thesis to explore the internal neural mechanism of individual differences in conflict adaptation, and found that DLPFC and PPC play an important role in conflict adaptation, and the individual differences in spontaneous brain activity and PPC structure of DLPFC will affect individual differences in conflict adaptation. The study of brain function of addicts also found that DLPFC and PPC of addicts were damaged in different degrees. Our research also found that the unconscious conflict adaptation of addicts is impaired, which may be related to the brain function impairment of addicts.

Question6: “This can help to explain the ability of addicts to suppress unconscious stimuli, and further reveal the relationship between the anti-interference ability and relapse behavior of heroin abstainers when facing drug-related stimuli.” I think the introduction could be more specific on this point.

Answer:Thanks for the expert's correction and reminder. Regarding this part, we have made some modifications in the introduction of the text, explaining the relationship between unconscious conflict adaptation and relapse (see the modification of the introduction for details).

Question7: So  these are long time drug users/addicts?

Answer:Thank you very much for the expert's reminder. The mean age of heroin abstainer group was 48.58 ± 4.53 years, the first drug use age was 30.24 ± 9.70 years, the mean duration of current abstinence was 8.42 ± 4.62months (range: 4–24 months). In terms of withdrawal duration, the subjects in this sample did not exactly fall into the category of long-term abstainers.

Question8: Impulsivity can be assessed from questionnaire, what does this new task add? How does this correlate with experimental results?

Answer:Special thanks to experts for their valuable opinions and corrections. Experts' opinions are very important for the rigor of our research. The BIS-11 scale is added to control the extra influence of individual impulsivity on the experimental results. By analyzing the difference of impulsivity between the two groups of subjects, the influence on the experimental results is significant, but not significant, so we exclude the influence of individual impulsivity on the experimental results.

Question9:The paper has  no analysis section-this makes the results difficult to interpret. There should be a statement of all he experimental hypotheses.

Answer:Thank you very much for your advice and guidance. Experts' opinions make our research clearer in presentation and presentation, and the analysis explanation and the statement explanation of research hypothesis have been added to the result part. we hypothesized that drug addicts had impaired unconscious conflict adaptation.

Question9: pre-test and post-test of the heroin abstainers group’s craving for drugs (t(37)= 0.66, p>.005. Was this anticipated? The whole results section is complicated and difficult to follow. Table 1 is not referred to in the text.

Answer:Thanks for the expert's correction and reminder. Before and after the test, addicts found that there was no significant difference in their craving for drugs, which was in line with the expectation of our experimental operation. Desire greatly affects the cognitive function of addicts, and it will also affect their task performance. Then, if there is no difference, it means that there is no or little influence of craving in the experimental process. In addition, if there is a change in craving, it is necessary to do some work to reduce craving, because it also involves experimental ethical issues. In order to present the results more clearly, we have revised the results, and Table 1 has been added again in the text. Thanks again for the guidance of the experts (see the results for details).

Question10:Again, what is the specifically new finding here?

Answer:Thank the experts for their corrections and reminders, and once again apologize to the experts for our unclear expression. We found that both addicts and control group have unconscious conflict adaptation effects, but compared with control group, the absolute value of conflict adaptation of addicts is larger, which indicates that heroin addicts have weaker unconscious conflict adaptation. That is to say, when heroin addicts face unconscious information interference, their anti-interference ability will also be reduced.

We have rewritten the conclusion: An arrow version meta-contrast masking task was used in the present study to explore unconscious conflict adaptation in heroin addicts. According to the findings, heroin users exhibited much lower conflict effects following incongruent trials than they did following congruent trials, indicating that an unconscious conflict adaptation effect had occurred. More significantly, heroin addicts' unconscious conflict adaptation effect was significantly smaller than that of the healthy control group, indicating that heroin addicts may have impaired unconscious conflict adaptation as a result of functional brain impairment brought on by long-term drug use.

Question11:Was the amount of conflict especially large?

Answer:Thank you very much for your advice and guidance. Based on the available studies I only found significant differences between conflict adaptation in the heroin withdrawal group and the normal group.

Question12:From here to 209. These results are interesting-what do they mean? What about the pre study measure of drug craving/impulsivity?

Answer:In order to more directly reflect the difference of unconscious conflict adaptation between the two groups of subjects, the conflict effects were calculated respectively. The independent sample T test was used to compare the conflict effects between heroin addicts and the control group. The results showed that there was a significant difference between the unconscious conflict adaptation of heroin addicts and that of the normal control group. The conflict adaptation of heroin addicts was larger than that of the control group, which indicated that the unconscious conflict adaptation of heroin addicts was damaged.

Before the start of the experiment, we used Visual Analogue Scale, VAS) to assess the craving scale (0-10), with 0 indicating no craving and 10 indicating strong craving. Then we used Barratt-11 Impulsivity Scale to assess the impulsivity level of the subjects. The higher the score of the scale, the more impulsive it is.

Question13: Discuss absolute value in more detail-for example, is there a threshold? Can we say that over a certain level is going to make withdrawal harder.

Answer:First of all, thank the experts for their suggestions and reminders. The expert's question is very good, but unfortunately, our current research can't give an answer, because our definitions of injuries are all produced by comparing with those of the control group (healthy people), and we don't judge whether there is a specific value for addicts' conflict adaptation injuries, but we are also considering conducting follow-up research to explore whether there is a threshold for this kind of conflict adaptation injuries of addicts.

Question14: “This may indicate that the conflict inhibition function of drug addicts is impaired due to long-term drug abuse, making addicts spend more time in solving conflict inhibition than health people[31]. In addition, in a study of electroencephalogram” does this deficit exist prior to drug or is it the result of drug use.

Answer:Special thanks to experts for their valuable opinions and corrections. In the past, most researchers thought that the cognitive control function of addicts was damaged after addiction, so we also thought that conflict adaptation damage did not exist before addiction.

 Question15: In the face of conflict, more cognitive resources need to be mobilized, and the control of irrelevant information interference was poor.” Not sure what this means. If the conflict is not conscious how can “more cognitive resources need to be mobilized”. What are these cognitive resources?

Answer:Thank you very much for your questions. The purpose of this study is to explore whether the conflict adaptation of addicts needs the participation of consciousness from the behavioral level. The results may also reflect the commonness and individuality of the unconscious and conscious conflict adaptation in addiction. The commonness lies in the fact that both unconscious conflicts and conscious conflicts can activate the adaptive control system of human beings, and the individuality lies in the difference of activation degree between unconscious conflicts and conscious conflicts.

Our understanding of the expert's question is as follows: compared with the situation without conflict, conflict information itself can be regarded as a new and different stimulus. When people face new and different stimuli, they may unconsciously or consciously allocate more cognitive resources to judge whether they pose a threat to themselves. However, with the continuous emergence of conflict information, people control it, and may slowly adapt to it, so as to devote their limited cognitive resources to more important things. In this process, conscious conflict control may be transformed into unconscious conflict control.

 Question16: Does this mean that it is depression rather than drug use that is related to poorer performance?

Answer:Thanks for the expert's question. In order to eliminate the influence of subjects' emotional state on the experimental results, we conducted a two-factor repeated measurement ANOVA of emotional state (pre-test, post-test) × types of subjects (addiction group, control group) after the end of the experiment. The results showed that the main effect of pre-test and post-test of heroin addiction group and normal control group was not significant . Therefore, we also ruled out the influence of emotional state changes before and after the experiment.

Question17:“cognitive control ability of unconscious conflict improved with abstinence time” Earlier text does not mention this?

Answer:Special thanks to experts for their valuable opinions and corrections. It is a controversial topic whether the cognitive function of an addict can recover with the withdrawal time. In order to make our research more accurate, the controversial argument has been revised (see the outlook revision section for details).

Question18:The conclusion could really draw out what new information this study provides

Answer:Thank you very much for your valuable advice and guidance. From the behavioral level, we explored whether the addicts' conflict adaptation needs conscious participation and whether this unconscious conflict adaptation also has damage. The results showed that the addicts' unconscious conflict adaptation exists and is accompanied by damage. This part has been revised again (see the revised part of the conclusion for details).

Reviewer 2 Report

I am afraid that the clarity of your reporting needed to be higher. Of fundamental importance is that a clearer explanation of what you meant by consistent trials after consistent trials, etc, was far from clear. For example, over how many trials was consistency or inconsistency assessed? These matters were fundamental to your results, so that without clarity here, it became impossible to understand what your reported results actually meant.

One other very important matter is ethical in nature. Despite your statement  concerning the Helsinki declaration, it appears that you were working with a group of compulsory patients. Therefore, you need to emphasise further that their participation was voluntary, and that steps were taken to avoid coercion to participate. On a related point, how and from where was the control group recruited?

There were several mistakes of English grammar in the article, and a greater use of tables, within the limits allowed by the journal would have aided clarity. Finally, I could not see any explanation as to why a reference list has been included as supplementary material.

Author Response

Reviewer2:

We are very grateful for the reviewers' comments and suggestions, and we also found that many of our first draft statements were not very clear, and we have rewritten the manuscript based on this. We hope that this manuscript will answer many of the reviewers' questions.

Question1: I am afraid that the clarity of your reporting needed to be higher. Of fundamental importance is that a clearer explanation of what you meant by consistent trials after consistent trials, etc, was far from clear. For example, over how many trials was consistency or inconsistency assessed? These matters were fundamental to your results, so that without clarity here, it became impossible to understand what your reported results actually meant.

Answer:Thank you very much for your valuable advice and guidance. Specifically, in the arrow Flanker task, the subjects were presented with a row of five arrows, and asked to respond to the target arrow in the middle and ignore the interfering arrows on both sides. Set two types of stimuli: consistent (C, interference arrow is consistent with target arrow) and inconsistent (I, interference arrow is inconsistent with target arrow). Usually, the subjects' response to consistent attempts is faster than that to inconsistent attempts, and the difference of the reaction time between the two attempts is the flanker interference effect. If the relationship between before and after trials is also included in the analysis, there will be four types of stimulation: CC (previously consistent and currently consistent), CI (previously consistent and currently inconsistent), IC (previously inconsistent and currently consistent) and II (previously inconsistent and currently inconsistent). The conflict adaptation effect means that the interference effect (II-IC) after inconsistent attempts is smaller than the interference effect (CI-CC) after consistent attempts, that is, (CI-CC)>(II-IC), and the size of conflict adaptation is expressed by [(CI-CC)-(II-IC)]. There are 7 Blocks in our formal experiment, and each block has 128 trials, of which the number of consistent and inconsistent trials is equal, with 64 trial each. The experimental process has been modified accordingly (see method section 2.4 experimental process for details).

Question2: One other very important matter is ethical in nature. Despite your statement  concerning the Helsinki declaration, it appears that you were working with a group of compulsory patients. Therefore, you need to emphasise further that their participation was voluntary, and that steps were taken to avoid coercion to participate. On a related point, how and from where was the control group recruited?

Answer:Special thanks to experts for their valuable opinions and corrections. Our experiment strictly abides by the experimental ethics from beginning to end. First of all, addicts are selected from many drug addicts in compulsory isolation and detoxification centers according to DSM standards. There is no compulsory participation of participants in the experiment. In fact, participants who do not meet the requirements will not be allowed to participate in the experiment, and they also have the right to refuse the experiment. Secondly, before the experiment, we have informed the participants about the content of the experiment and their right to quit halfway. However, because we are worried about the bias effect of the experimenters, we have concealed some experimental purposes. At the end of the experiment, we all told the purpose of the experiment separately. They also signed the informed consent form. Finally, our research group has cooperated with the drug rehabilitation center for ten years, and we will also give them some group psychological counseling and other activities, so they have a certain degree of acceptance of psychology. Generally speaking, although the participants do come from compulsory isolation drug rehabilitation centers, there are no coercion or ethical problems.

Question3: There were several mistakes of English grammar in the article, and a greater use of tables, within the limits allowed by the journal would have aided clarity. Finally, I could not see any explanation as to why a reference list has been included as supplementary material.

Answer:Thank you for your valuable opinions. We have fully adopted the first opinion of the experts. Specifically, we used tables to present the experimental results in the part of experimental results, and polished the language again in this article. For references, we have attached them to the text again. For the previous practice of placing references in appendix materials, we mistakenly thought that the text had a typesetting word limit, so we submitted the texts of references to your department separately.
